# Personalised Medicine for Colorectal Cancer Using Mechanism-Based Machine Learning Models

**DOI:** 10.3390/ijms22189970

**Published:** 2021-09-15

**Authors:** Annabelle Nwaokorie, Dirk Fey

**Affiliations:** Systems Biology Ireland, School of Medicine, University College Dublin, Belfield, Dublin 4, Ireland; annabelle.nwaokorie@ucdconnect.ie

**Keywords:** cancer, colorectal cancer, signal transduction networks, pathways, event-free survival, biomarkers, WNT pathway, targeted therapy

## Abstract

Gaining insight into the mechanisms of signal transduction networks (STNs) by using critical features from patient-specific mathematical models can improve patient stratification and help to identify potential drug targets. To achieve this, these models should focus on the critical STNs for each cancer, include prognostic genes and proteins, and correctly predict patient-specific differences in STN activity. Focussing on colorectal cancer and the WNT STN, we used mechanism-based machine learning models to identify genes and proteins with significant associations to event-free patient survival and predictive power for explaining patient-specific differences of STN activity. First, we identified the WNT pathway as the most significant pathway associated with event-free survival. Second, we built linear-regression models that incorporated both genes and proteins from established mechanistic models in the literature and novel genes with significant associations to event-free patient survival. Data from The Cancer Genome Atlas and Clinical Proteomic Tumour Analysis Consortium were used, and patient-specific STN activity scores were computed using PROGENy. Three linear regression models were built, based on; (1) the gene-set of a state-of-the-art mechanistic model in the literature, (2) novel genes identified, and (3) novel proteins identified. The novel genes and proteins were genes and proteins of the extant WNT pathway whose expression was significantly associated with event-free survival. The results show that the predictive power of a model that incorporated novel event-free associated genes is better compared to a model focussing on the genes of a current state-of-the-art mechanistic model. Several significant genes that should be integrated into future mechanistic models of the WNT pathway are DVL3, FZD5, RAC1, ROCK2, GSK3B, CTB2, CBT1, and PRKCA. Thus, the study demonstrates that using mechanistic information in combination with machine learning can identify novel features (genes and proteins) that are important for explaining the STN heterogeneity between patients and their association to clinical outcomes.

## 1. Introduction

Colorectal cancer (CRC) is the second leading cause of cancer-related death worldwide [1,2,3]. From the four global consensus molecular subtypes (CMS) of CRC, CMS1–4, no specific drugs have been identified that target a specific CMS [4]. However, there are multiple studies showing that some CMS subtypes benefit from specific therapeutic regimens [5,6,7]. For CMS2 and 3, both of which are poorly immunogenic in comparison to CMS1 and 4, the only associated treatment is toxic chemotherapy, which results in poor patient prognosis [2,8]. Biomarkers are foundations of clinical and personalised medicine; despite being vital tools in the area of clinical diagnostics, many are based on single time-point measurements and lack the dynamic information that is needed to follow diseases and therapy [9]. Therefore, we hypothesised that by integrating tumour profiling data with dynamic information about signal transduction networks (STNs), mechanism-based machine learning models could aid in the prediction of patient survival and response to therapy, and overall provide insight into the disease and drug-response mechanisms and reveal potential drug targets and novel biomarkers.

STNs are extremely adjustable and dynamic [10]. To better understand these STNs in the context of CRC, knowledge about how active these STNs are and how this activity varies between patients and relates to patient outcomes is desired. To date, numerous CRC STNs have been identified, including TGF-β (transforming growth factor beta), PI3K/Akt (phosphatidylinositol 3-kinase/protein kinase B), TP53 (tumour protein P53), MAPK (microtubule-associated protein kinase), Apoptosis, Cell Cycle (cell-division cycle), mTOR (mammalian target of rapamycin), Notch, and WNT (Wingless-related integration site) [3,11]. Table 1 portrays these critical CRC STNs and their associated functions. This study analyses only five of these critical pathways, including TGF-β, PI3K/Akt, TP53, MAPK, and WNT, because PROGENy (pathway responsive genes for activity inference) is not available for Notch, Apoptosis, mTOR, and Cell Cycle.

STNs can be mapped to their most frequent mutations. Interestingly, APC (adenomatous polyposis coli) negatively regulates WNT activity since it is an integral part of the destruction complex, which targets ß-catenin for degradation. Mutations in APC thus lead to hyperactive WNT signalling [18]. Across all CRC STNs, the role of WNT signalling in carcinogenesis has most notably been described for CRC [24]. Roughly all CRCs present hyperactivation of the WNT pathway, which in many cases is believed to be the initiating and driving event [3,25,26]. Despite this, most mechanistic models of the canonical WNT signalling pathway (β-catenin-dependent) are based on solely the intracellular steps, neglecting the extracellular steps [27,28,29,30,31,32]. Classically, the term “canonical” for this pathway refers to the pathway components, which lead to the stabilisation of β-catenin in response to WNT ligands. Consequently, the analysis for this study was restricted to this specific WNT sub-pathway, i.e., canonical. In contrast to most studies, Kogan et al. developed a flexible framework for identifying potential drug targets in the WNT STN. This mechanistic model was based on the initial steps of the activation solely and analysed the effects of two extracellular inhibitors, DKK (Dickkopf) and sFRP (secreted frizzled-related protein) [27]. This validated mathematical model, which predicts effective combinational therapy by sFRP and DKK, is the most recent WNT STN model to date and, interestingly has never been validated in human cell lines. Overall, it is not instinctively apparent which components of the WNT STN are best to target for therapeutic intervention or how such interventions should be designed in order to achieve the best clinical outcomes [27]. However, it has emerged that in recent years, using patient-specific mathematical models can improve patient-stratification and help to identify potential drug targets and largely improve patient response to therapy [9,10,33]. Additionally, combining mechanism-based machine learning models of disease dynamics has been proven to enhance the development of novel disease interventions [34]. Concretely, mechanism-based machine learning models are firmly based upon the behaviour and critical features of mechanistic signal transduction networks of interest, whilst formed by classic linear regression. Adapting this, a schematic of the approach is shown in Figure 1.

In the present study, we developed adjustable linear regression machine learning models of the activation of the canonical WNT signalling pathway and incorporated genes and proteins that were significantly associated with event-free patient survival in CRC. The term “adjustable” in such models represents the capability to vary (include/remove) features with ease to determine which features are most important. The results revealed that the machine learning models can not only help to understand the behaviour of the initial steps of this complex network by identifying novel features not yet included in mechanistic models but also interpret its behaviour by relating pathway activity to clinical outcomes on the level of individual patients.

## 2. Results

### 2.1. Determining Significant Genes and Proteins in CRC: A Kaplan–Meier Survival Analysis

In line with the literature, five signalling pathways critical in CRC were selected to analyse, from several network databases, including WikiPathways, Gene Set Enrichment Analysis (GSEA), and KEGG [14,21,22,37,38]. These five signalling pathways comprise WNT, PI3K-Akt, TP53, MAPK, and TGF-Beta. As mentioned above, the motivation behind choosing these five pathways is due to the limited data available to perform a PROGENy analysis. The five STNs analysed are the only CRC STNs available on PROGENy. To determine which genes and proteins are significant in CRC, components of each pathway were used to integrate data from the listed resources. In total, 1156 features were analysed across all pathways. The correlation between patient event-free survival and the expression of proteins and genes from a Kaplan–Meier and log-rank test resulted in 464 significant features. In total, 389 genes and 75 proteins were found to be significant across the eight major CRC signalling pathways. A concrete list of all significant genes and proteins is available in the supplemental data. The associated hazard ratios and *p*-values of each significant feature are denoted below in Table 2 and Table 3. All significant genes were obtained from the transcriptomic TCGA (The Cancer Genome Atlas), Firehose legacy dataset, with a cohort of 329/636 patients. The 329/636 TCGA patients selected were solely from the transcriptomic, mRNA (messenger ribonucleic acid) expression dataset; all patients had the necessary clinical data available. Similarly, all significant proteins were obtained from the proteomics TCGA Firehose legacy dataset, a cohort of 73 patients [39,40]. The associated clinical information can be found in the supplemental data.

#### 2.1.1. Significant Genes from RNA Sequencing

The total number of genes whose RNA expression correlated with event-free patient survival in this cohort was 389 of 872 genes across all CRC signalling pathways considered. For most pathways, including the WNT pathway, approximately half of the genes were significant using a fold change and *p*-value cut-off of 0.5 and 0.05, respectively.

Next, we wanted to obtain a more comprehensive picture of the significant genes within a pathway. Using the WNT pathway as an exemplar, the volcano plot in Figure 2 shows the *p*-values of the significant genes in the *y*-axis against their $\log_{2}$ fold change on the *x*-axis. Genes above the dashed threshold line, to the right, DVL3 (dishevelled segment polarity protein 3), VANGL2 (vang-like protein 2), CER1 (cerberus 1), and TCF7L1 (transcription factor 7-like 1), etc., are found to be significant, with the lowest *p*-values and greatest positive hazard-ratio fold-changes. These genes were associated with increased risk. Conversely, genes to the left above the threshold line, including MAPK8, have a negative hazard-ratio fold change and were thus associated with decreased risk. Overall, 15 genes were found to be significant negative markers, and the remaining 39 genes were significant positive markers determined for the WNT signalling pathway.

Plotting the event-free survival Kaplan–Meier curve for DVL3, the most significant gene in the WNT signalling pathway, resulted in a $\log_{2}$ hazard ratio of 1.49 and a 95% confidence interval between 0.69 and 2.29 (Figure 3). Kaplan–Meier scanning to determine the optimal cut-off divided the patients into high (*n* = 26) and low (*n* = 303) groups by scanning the group size from 10–90 to 90–10 percent splits, where 10–90 means that 10% of the patients were in the low group and 90% of the patients in the high group and calculating the *p*-value for the overall event-free survival difference between the groups using a log-rank test with Yates’ correction. A high expression of DVL3 resulted in poor patient survival, where a 2-year event-free survival equals 38% in the high group (*n* = 26). The Kaplan–Meier curves for the most significant gene found for each CRC STN is shown in Figure 3. 

#### 2.1.2. Significant Proteins from Proteomics

Across all of the CRC signalling pathways considered, 75 of 284 proteins exhibited expressions that correlated with event-free patient survival. For each pathway, about 4–10% of the pathway proteins exhibited an association with event-free survival. The top two significant proteins across each CRC signalling pathway are shown in Table 3. PRKCA (protein kinase c alpha) and ROCK2 (rho associated coiled-coil containing protein kinase 2) were found to be the most correlated to the patient’s survival in the WNT signalling pathway. PRKCA was found to be the most significant protein with an associated hazard ratio of 0.1557 and a *p*-value of $1.61\times 10^{-3}$. The signalling pathway PI3K-Akt has the greatest number of significant proteins associated, 29 proteins in total. 

Six significant proteins within the WNT signalling pathway were found from the proteomics dataset, including PRKCA, ROCK2, CSNK2A1 (casein kinase 2 alpha 1), LRP1 (low-density lipoprotein receptor-related protein 1), CSNK1A1 (casein kinase 1 alpha 1), and GPC4 (glypican 4). This is represented by the volcano plot in Figure 4. In total, 92 proteins in this pathway could not be analysed because of missing values. Significant proteins, ROCK2 and CSNK2A1, have a positive hazard ratio with a $\log_{2}$ fold change above 1.5. All other significant proteins have a low-risk association with hazard ratios below −1.5. In summary, six proteins were found to be significant, two with an increased risk and four with decreased risk. 

Plotting the event-free survival Kaplan–Meier curve for PRKCA, the most significant protein in the WNT signalling pathway, resulted in a $\log_{2}$ hazard ratio of −2.68, a confidence interval between −4.35 and −1.02, and a *p*-value of $1.61\times 10^{-3}$. From Kaplan–Meier scanning, 55 patients were in the high expression group, and 8 were in the low expression group. High expression of PRKCA resulted in improved patient survival, with 2-year event-free survival of about 90% in the high group (*n* = 55). The Kaplan–Meier curves for the most significant proteins found for each CRC STN are shown in Figure 5.

#### 2.1.3. Summary of the Event-Free Survival Analysis Results

Across five critical CRC signalling pathways, the results from the WNT signalling pathway introduced several significant genes and proteins that have not yet been accounted for in current WNT mechanistic models. Therefore, the results of this section reinforce the need to propose a strategy to incorporate them into current WNT mechanistic models. To summarise, Table 4 is composed of all 53 significant genes and 6 significant proteins found in the WNT signalling pathway. These results were from the TCGA Legacy, the transcriptomic dataset of 329 CRC patients, and the proteomic from mass spectroscopy dataset of 73 patients. Interestingly, some significant genes (DVL3, DKK1, SFPR2, WNT3A, WNT3, SFPR1, and GSK3B) are found in published mechanistic models of the WNT pathway [27,28,29,30,31,32]. The remaining 52 significant features found in this study are not found in these mechanistic models. Thus, we have identified many genes that were associated with event-free survival and that constitute prime candidates to improve current WNT mechanistic models. These improved models could then be used for patient stratification and predicting response to therapy [9,33].

### 2.2. Predicting Pathway Responsive Genes for Activity Interference from Gene Expression: A PROGENy Analysis 

Having identified the significant genes and proteins within the pathways, we now sought to obtain activity scores for each pathway. The purpose was two-fold. Firstly, to identify pathway activities that correlate with event-free patient survival; and secondly, to relate the genes and proteins expression of a pathway to its activity score. The RStudio package function PROGENy was used to obtain pathway scores from the RNA sequencing data of the TCGA legacy dataset. Albeit PROGENy’s composition of 14-cancer relevant pathways, specifically for this study, only the CRC pathways were analysed in depth. The only available CRC pathways in PROGENy were PI3K, MAPK, TGF-Beta, WNT, and p53. 

The PROGENy activity scores for each CRC signalling pathway are shown in Figure 6. Although the TGF-Beta and PI3K appear to have an inverse relationship, most pathways exhibit a cluster of high activity scores in a small subset of patients. However, specific activity patterns are difficult to discern. 

#### Kaplan–Meier Survival Analysis on PROGENy Activity Scores

To investigate the correlation between the PROGENy pathway activity scores and patient event-free survival, a Kaplan–Meier analysis and log-rank test were used. The activities of the WNT, PI3K, TGF-Beta, and MAPK pathways were all significantly correlated with event-free survival (Table 5). In particular, we found that activity of the WNT STN exhibited the strongest association with event-free patient survival (hazard ratio and *p*-value of 1.9731 and 0.0013, respectively).

The event-free survival Kaplan–Meier curves for the CRC STNs MAPK, TGF-Beta, PI3K, p53, and WNT, associated with patient survival through PROGENy activity scores, are shown in Figure 7. The STNs MAPK, TGF-Beta, and WNT followed a similar trend, where a high pathway activation was associated with shorter event-free survival. From the literature, this is expected considering that both pathways, MAPK and WNT, drive cell proliferation. Activation of the WNT pathway in CRC increases the levels of β-catenin within the cytosol, causing it to further travel into the nucleus and express WNT target genes, including genes that control the cell cycle [41]. For the WNT signalling pathway, the $\log_{2}$ hazard ratio was 0.98 with a 95% confidence interval between 0.38 and 1.58 and a *p*-value of $1.3\times 10^{-3}$. The high group characterised by high WNT activity consisted of 108 patients and the low group of 221 patients. Similarly, the results for TGF-Beta are in line with its immunosuppressive function. Conversely, for the other STNs, p53 and PI3K, a high pathway activity was associated with longer event-free survival. The p53 result is in line with its known functions as a tumour-suppressor. The PI3K result is counterintuitive considering that the PI3K/AKT pathway is a classical survival signalling pathway but might be a confounding factor explained by “side-effect” activation due to crosstalk.

### 2.3. Developing Linear Regression Machine Learning Models

Having ascertained the significant genes, proteins, and pathways, we next sought to determine how well the expression of genes and proteins of a pathway can predict its activity scores. The results should give valuable insight into which candidate genes and proteins should be prioritised in follow-up studies focusing on constructing patient-specific mechanistic models with significant value. The WNT STN was the most significant pathway (Table 5, Figure 7); thus, we focused solely on it. Because there was a limited overlap between the mRNA and proteomics data in the TCGA legacy dataset (only five patients had data for both), we focused the analysis on the CPTAC (Clinical Proteomic Tumor Analysis Consortium) dataset. Both mRNA and proteomics data were available for the entire CPTAC cohort of 79 patients. 

In total, three machine learning models were developed to predict the PROGENY pathway scores using different features as inputs. The features were the genes and proteins of the WNT pathway and their associated gene or protein expression values. To minimise the risk of overfitting, 10-fold cross-validation was used to build each model. The predictive power of each model was judged based on the root mean squared error (RMSE), the correlation coefficient (R), and the *p*-value. A *p*-value less than 0.05 was taken as significant for this field of study. The RMSE is an appropriate measure of predictive power for this study because model complexity was not an issue, and RMSE measures the differences between predicted and actual PROGENy values. There is no absolute threshold; however, lower values indicate a better overall fit of how accurately the model predicts the response. An acceptable range for the RSME values is difficult to define, but Model 1, consisting of the genes of the current mechanistic model, can act as a baseline benchmark. Notably, the predicted values of Model 2 show a much better correlation with the true values and a lower RMSE value compared to Model 1. (Figure 8B); thus, demonstrating a much better fit. We focus our analysis on three models:Model 1: Features are eleven genes taken from the most recent WNT mechanistic model developed by Kogan et al.: APC, AXIN1, CTNNB1, DKK2, DVL3, GSK3B, LRP6, SFRP1, SFRP2, and WNT3.Model 2: Features are the mRNA expression of nine genes: seven features selected using LASSO and the two most significant genes (DVL3, VANGL2): DKK3, FZD5, NKD1, NOTUM (notum, palmitoleoyl-protein carboxylesterase), WNT11, PRKCA, and ROCK2.Model 3: Features are the protein expression of seven proteins; five identified using LASSO and the two most significant proteins (PRKCA, ROCK2): CTBP1 (c-terminal binding protein 1), CTBP2 (c-terminal binding protein 2), GPC4, PLCB4 (phospholipase c beta 4), and RAC1 (ras-related C3 botulinum toxin substrate 1).

Model 1 was developed based on the mechanistic WNT signalling pathway. The selected features were the eleven genes of the mechanistic models in the literature [27,28,29,30,31,32], using the mRNA expression data from the CPTAC dataset (Table 6). The gene DKK1 is typically found in the mechanistic WNT pathway; however, it was not available in the RNA CPTAC dataset; therefore, its paralogue, DKK2, replaced it [42]. Models 2 and 3 were developed using systematic feature selection based on the entire set of 119 WNT pathway genes. LASSO regression was used to identify the features based on the mRNA expression data (Model 2) or protein expression data (Model 3). To complete the model, the two most significant genes (Model 2) or proteins (Model 3) were also included as features. LASSO regression on the mRNA data of the 119 WNT network genes selected seven genes as features. Thus, Model 2 consisted of nine features: seven identified by LASSO plus the two most significant genes. LASSO regression on the protein data of the 27 proteins of the WNT network for which data were available selected five proteins as features. Thus, Model 3 consisted of seven features: five identified by LASSO plus the two most significant proteins. 

How well can the mechanism-based, mRNA-based, and protein-based models predict the pathway scores? Model 1, based on the genes of the mechanistic models, serves as a benchmark and achieved a reasonable predictive power with a Spearman correlation coefficient of 0.5241, a *p*-value of $7.12\times 10^{-7}$, and an RMSE of 1.0167. Model 2, based on LASSO identified genes from the canonical WNT pathway, resulting in a greater predictive power, compared to Model 1, with a Spearman correlation coefficient of 0.8367, a *p*-value of $7.88\times 10^{-22}$, and an RMSE of 0.6416. Despite the smaller number of input features for Model 2, compared to Model 1, Model 2 evidentially has greater predictive power, shown by the higher *p*-value, correlation coefficient, and lower RMSE. Similarly to Model 2, Model 3 used LASSO to identify proteins from the canonical WNT pathway. In total, data for only 27 proteins were available due to missing data and imputation (see materials and methods). Model 3 is the least predictive model. The associated Spearman correlation coefficient, RMSE, and *p*-value were 0.4095, 1.0509, and $1.78\times 10^{-4}$, respectively. Most features used in these machine learning models have nonzero regression coefficients (Figure 8D–F).

### 2.4. Summary

By incorporating the results from the event-free survival analysis, the PROGENy analysis, and the development of the linear regression machine learning models together, it is apparent that each finding paves the way to a valuable tool for predicting the heterogeneity of the WNT STN activity on an individual patient level. Concretely, each finding from each analysis builds sequentially upon another. The event-free survival analysis brings forth critical and novel features for the WNT pathway that have not yet been accounted for mechanistically. Interesting, from the PROGENy analysis, we found that the WNT signalling pathway was the most significant pathway associated with event-free patient survival. This finding solidifies the idea that there is a relationship between this pathway and patient survival that should be continued to be researched mechanistically. Finally, the motivation for using linear regression machine learning was particularly because it is an appropriate tool to predict WNT pathway activity from specified features (i.e., prognostic genes and proteins) on an induvial patient level. Precisely, the machine learning models additionally identified several features that were not only significantly associated with event-free patient survival, but also important for predicting the patient-specific WNT pathway activity scores. Thus, by incorporating such features into an updated mechanistic model of the canonical WNT STN activation, one would expect to better understand the patient-specific differences in the control of pathway activity, but also to relate pathway activity to clinical outcomes on the level of individual patients.

## 3. Discussion

Our experimental workflow employed numerous Kaplan–Meier survival scans, a PROGENy analysis, and developed three machine learning models based on mechanistic WNT pathway models and LASSO identified genes and proteins of the extant WNT network. The models can serve as a platform for improving current mechanistic models and WNT pathway activity-based patient stratification. Conversely, all models of the WNT signalling pathway to date have focussed on a few genes and proteins of the core pathway [27,28,29,30,31,32]. Building upon this work that describes the initial steps of WNT pathway activation, our work builds upon this by identifying new features and regression models that describe the heterogeneity of the WNT pathway activity scores across patients from two CRC cohorts [27]. We have also shown that the activity scores of the WNT pathway were the most significant in terms of the association to event-free patient survival compared to four other key pathways of CRC; the MAPK, PI3K, TGF-Beta, and p53 pathways. 

### 3.1. Event-Free Survival Analysis

This study determined novel genes and proteins significantly associated with event-free survival in CRC across the major five CRC signalling pathways, WNT, PI3K-Akt, TP53, MAPK, and TGF-Beta. In total, 389 genes and 75 proteins were found to be significant across all signalling pathways from a sum of 872 genes and 284 proteins. A total of 59 of these features were significant in the WNT signalling pathway, and interestingly, only seven of these significant features, DVL3, DKK1, SFPR2, WNT3A, WNT3, SFPR1, and GSK3B, are found within previous mechanistic WNT models [27,28,29,30,31,32]. Thus, in order to build new and improved mechanistic models of the WNT pathway with the potential to fulfil the promise of precision medicine, future modelling should focus on these significant genes and proteins.

For the WNT signalling pathway, the number of significant genes determined from RNA sequencing was 53. The top two significant genes found were DVL3 and VANGL2. The most significant gene was DVL3, with a corresponding hazard ratio of 2.8158 and a *p*-value of $\;2.58\times 10^{-4}$. WNT ligands bind to FZD receptors and the LRP5/6 co-receptor, which leads to the recruitment of cytosolic DVL3, then relays this signal to downstream signalling events that result in the translocation of ß-catenin to the nucleus and target gene expression. Thus, DVL is recruited by the receptor Frizzled and prevents the fundamental destruction of cytosolic β-catenin [43]. Previous studies demonstrated that a high expression of DVL3 in CRC acts as an unfavourable prognostic marker [44]. Additionally, Zhao et al. suggest that DVL3 is a key regulator in CRC chemoresistance and targeting it may be a potential strategy for CRC therapy [45]. Our study solidifies this further, as shown in the Kaplan–Meier curve in Figure 3, in which a high expression of DVL3 was associated with a lower event-free patient survival. VANGL2, the second most significant gene in the WNT signalling pathway, has a hazard ratio and *p*-value of 2.7692 and $1.08\times 10^{-3}$, respectively. Studies show that VANGL2 can be activated by WNT through Frizzled receptors [46]. Furthermore, within the WNT pathway, six proteins were significantly associated with event-free patient survival. The most significant protein was PRKCA. High PRKCA protein expression correlated with longer event-free survival (Figure 5). Interestingly, the PRKCA protein was also important for explaining the WNT pathway activity scores in Model 3, with a significant negative regression coefficient (Figure 8), which is in line with the known function of PRKCA, which is to inhibit WNT signalling via several mechanisms, including the phosphorylation of β-catenin, and enhance CRC cell death, concretely fitting to the survival data presented in Figure 5 [47]. Together, these results suggest a model in which PRKCA exerts its positive effects on patient survival via repressing the WNT activity.

Finally, the results for the WNT pathway analysis identified several significant genes and proteins that were associated with event-free survival but have not yet been accounted for in current WNT mechanistic models. Most, if not all, models to date are based on a few intracellular and extracellular components of the core WNT pathway [27,28,29,30,31,32]. Focusing on describing the general mechanistic details of WNT pathway activation, these models were not intended to model patient specific differences and describe the WNT signalling heterogeneity across patients. In contrast, our analysis and machine learning models aimed at identifying genes and proteins with significant event-free survival associations and describing patient-specific differences of WNT pathway activation. In total, the remaining 52 significant features found in this study are not found in these mechanistic models. Thus, we have identified many genes that were associated with event-free survival and that constitute prime candidates to improve current WNT mechanistic models. These improved models could then be used for patient stratification and predicting response to therapy [9,33]. 

### 3.2. A PROGENy Analysis

The second part of this study consisted of a PROGENy pathway activity score analysis. There were several reasons why PROGENy and not classical GSEA was performed. One major reason PROGENy does not require predefined groups for comparisons, unlike GSEA, which determines whether a defined set of genes shows statistically significant, concurring differences between two groups, for example, normal and tumour tissue [14,48]. PROGENy, on the other hand, predicts specific pathway activity scores for each individual patient; thus, allowing us to build models that can explain these scores [35,49].

An advantage of the PROGENy analysis is that it reveals patient-specific differences, showing that different pathways are active in different patients (Figure 6). Interestingly, the PROGENy analysis did not reveal a striking hyperactivation of the WNT pathway (Figure 6); most values were found to be between 0 and 0.5 despite it being the main driving event in tumorigeneses within most CRC patients. This finding is explained by the fact that the pathway analysis is a differential analysis. PROGENy results are *z*-scores quantifying standardised (mean normalised and scaled) relative differences, yet all patients might have high WNT activity in absolute terms.

The results correlating the PROGENy pathway activity scores to patient event-free survival show that the WNT, PI3K, TGF-Beta, and MAPK pathways, when activated, were significantly associated with event-free patient survival. It is very interesting to note that the WNT signalling pathway was the most significant pathway associated with event-free patient survival. These results are in line with the literature, where high WNT pathway activation was associated with shorter survival [27,28,29,30,31,32]. The activation of the WNT pathway in CRC increases the levels of β-catenin within the cytosol, causing it to translocate into the nucleus and express WNT target genes that drive cell-proliferation [41]. How can the patient-specific differences of the WNT pathway activity be explained?

### 3.3. Development of Linear Regression Models

The final part of this study developed three linear regression machine learning models with the aim to predict the PROGENy WNT pathway activity scores. Current mechanistic models describe the pathway components and provide deterministic insights into how the WNT pathway is activated but neglect patient-specific differences [27,28,29,30,31,32]. Thus, combining machine learning that focuses on patient-specific differences with mechanistic models has great potential for building explainable machine learning models. 

Overall, Model 2 was found as the most predictive model. In comparison, the RMSE of Model 2 was 0.3751 times smaller than Model 1. Five features were identified that were both prognostic of event-free patient survival and predictive of WNT pathway activity, including DVL3, PRKCA, VANGL2, GPC4, and ROCK2. Of these, DVL3 was the only feature found in the mechanistic model, Model 1. The finding also highlights a lack of features in current mechanistic models that can explain patient-specific differences. Concretely, solidifying the need to propose such a strategy to incorporate them. In particular, our work identified the four overlapping features above (PRKCA, VANGL2, GPC4, and ROCK2) that should be the focus of mechanistic modelling in the future. Machine learning, Model 2, accounting for such features, outperformed the current mechanistic model and can accurately predict the WNT pathway activity scores.

Interestingly, the protein PRKCA was found to be both significantly associated with event-free survival and a significant predictive feature of the WNT pathway activity score (Table 6). The results in Figure 8 show that most features used in the machine learning models have nonzero regression coefficients, in particular: DVL3, FZD5, RAC1, ROCK2, GSK3B, CTB2, CBT1, and PRKCA. These features provide valuable WNT network nodes to be included in future mechanistic models of the WNT network. The presented machine learning models incorporate patient specific significant features that describe pathway activity and can thus be termed “explainable”. Despite this, these prognostic features have not yet been accounted for in current WNT models. The finding that a LASSO-based machine learning model (Model 2) identified new features outperforms the current mechanistic model, and can accurately predict the WNT pathway activity scores, highlights the need to improve current mechanistic models by incorporating these features. 

## 4. Conclusions

In conclusion, we developed machine learning models of the activity of the canonical WNT signalling pathway and incorporated significant genes and proteins that are associated with event-free patient survival. This model is a valuable tool for predicting the heterogeneity of WNT pathway activity on an individual patient level. In this study, we identified several features (DVL3, FZD5, RAC1, ROCK2, GSK3B, CTB2, CBT1, and PRKCA) that were not only significantly associated with event-free patient survival but also important for predicting the patient-specific WNT pathway activity scores. We expect that integrating these features into mechanistic models of WNT pathway activation in the future will not only help to better understand the patient-specific differences in the control of pathway activity but also to relate pathway activity to clinical outcomes on the level of individual patients.

## 5. Materials and Methods

### 5.1. Data Acquisition from the Literature

All data used throughout this study was open-source data. The first datasets were from the 2016 TCGA Colorectal Adenocarcinoma, GDAC Firehose Legacy study, previously known as TCGA provisional. The datasets used for this study included Genomic from RNA sequencing (20,532 genes), proteomic from mass spectrometry (5562 proteins), and the associated clinical dataset [39,40]. The TCGA legacy datasets, due to the minimum overlap between patients, were only used to determine prognostic genes and proteins within such patients. Pre-processing across the datasets was performed before the survival analyses were complete. Across all datasets, patients who did not have the associated censoring status or disease-free survival months available were removed. The second set of data was from the 2019 prospective CPTAC-COAD colon adenocarcinoma studies, genomic from RNA sequencing (13,482 genes), proteomic from mass spectrometry (6422 proteins), and clinical datasets were all downloaded from LinkedOmics [39,40,50,51]. Minor pre-processing across the CPTAC datasets was performed before the following analyses were complete. The pre-processing included analysing the datasets, RNA, proteomics, and clinical to ensure that the same patients were overlapping across all and, if not, removing these patients. In total, 79 patients remained across all datasets. Additionally, for the CPTAC proteomics dataset, several NAN values were present. These values were imputed for the machine learning LASSO regression analysis of the WNT STN. This was achieved in the application Perseus using a low-shifted distribution of width 0.3 and a downshift of 1.8 across the total matrix [52]. Furthermore, proteins with data points of 10% or less were removed. In total, 27/39 proteins in the WNT STN remained after imputation. All datasets analysed and pre-processed for this study can be found in the Appendix A; consequently, Table 7 and Figure 9 represent key metrics and pre-processing steps for all datasets used.

The five critical CRC STNs used in this study stemmed from the literature [3,11,23]. The network databases WikiPathways, KEGG, and GSEA, were used to create a list of gene-sets for each pathway including WNT, PI3K-Akt, TP53, MAPK, and TGF-Beta [21,22,37,38,53,54,55,56,57,58,59,60,61,62]. Each gene-set consisted of every gene listed for each homo-sapiens STN pathway on WikiPathways. These gene-sets were then used as input scripts to determine which genes and proteins are prognostic and correlate to patient event-free survival. All gene sets are found in the Appendix A.

### 5.2. Kaplan–Meier Survival Analysis

The Kaplan–Meier survival analysis was simulated three times, the first to find the prognostic genes from RNA sequencing, the second to find the prognostic proteins from proteomics, both using the TCGA legacy datasets, and finally to determine the associations between PROGENy pathway activity scores and event-free survival using the RNA CPTAC dataset. The optimum cut-off for stratifying the patient populations into low and high groups was identified by scanning the group sizes from 10–90 to 90–10 percent splits, where 10–90 means that 10% of the patients were in the low group and 90% of the patients were in the high group and calculating the *p*-value for the overall event-free survival difference between the groups using a log-rank test with Yates’ correction. The cut-offs were based on the TCGA legacy datasets: RNA sequencing (329 patients) and proteomics (73 patients), and the CPTAC RNA dataset (79 patients). The inputs for the first two simulations were based on the gene or protein sets for each CRC STN identified above. The output was a Kaplan–Meier curve for the gene or protein expressed, indicating the number of patients in the high expression or low expression group with the corresponding statistical values apparent. The inputs for the final simulation were the PROGENy CRC pathway activity scores, and the output was 5 Kaplan–Meier curves for each CRC STN. The number of patients in the high activity or low activity group was identified with the corresponding statistical values. All statistical computations and Kaplan–Meier analyses were performed in MATLAB (version R2020b Update 5 (9.9.0. 1592791), The MathWorks, Inc., Natick, MA, USA) using the statistics toolbox and the log-rank (www.mathworks.com/matlabcentral/fileexchange/22317-logrank (accessed on 7 April 2021)) and kmplot (www.mathworks.com/matlabcentral/fileexchange/22293-kmplot (accessed on 7 April 2021)) functions from the MATLAB (version R2020b Update 5 (9.9.0. 1592791), The MathWorks, Inc., Natick, MA, USA) file exchange [63].

### 5.3. PROGENy Analysis

PROGENy is a machine learning based tool installed from Bioconductor as an RStudio package (RStudio Team (2020). RStudio: Integrated Development for R. RStudio, PBC, Boston, MA URL http://www.rstudio.com/) [35,49]. A PROGENy analysis was performed on the TCGA RNA sequencing legacy dataset to find a common core of CRC pathway activities. Despite PROGENy’s composition of 14-cancer relevant pathways, specifically for this study, only the CRC pathways were analysed in depth [35,49]. The available CRC pathways using PROGENy were PI3K, MAPK, TGF-Beta, WNT, and p53. The associated PROGENy CRC pathway activity scores for the entire TCGA legacy patient cohort is found in the Appendix A. PROGENy (version 1.12.0) was used in the study [35,49].

### 5.4. Developing Linear Regression Machine Learning Models

The three linear regression machine learning models developed in the study were performed in MATLAB using the regression learner app [64]. The three trained models were a variety of linear, stepwise linear, and robust linear regression models. Each model consisted of 79 observations, i.e., the entire CPTAC patient cohort. Cross-validation of 10-fold was used for each model. All models were developed solely from the CPTAC datasets from RNA and proteomics. The selected input features for the models were obtained through LASSO regression of both datasets. For both LASSO regression analyses, the lambda value with minimal mean squared error plus one standard deviation was applied. A threshold (*p*-value) of 0.05 was used to determine if a feature was significant. For the proteomics dataset, this resulted in 5/27 features (proteins) found to be significant, including CTBP1, CTBP2, PLCB4, PRKCA, and RAC1. Similarly, for the RNA sequencing dataset, 5/89 features (genes) were significant, DKK3, FZD5, NKD1, NOTUM, and WNT11. All associated datasets are found within the Appendix A.

## Figures and Tables

**Figure 1 ijms-22-09970-f001:**
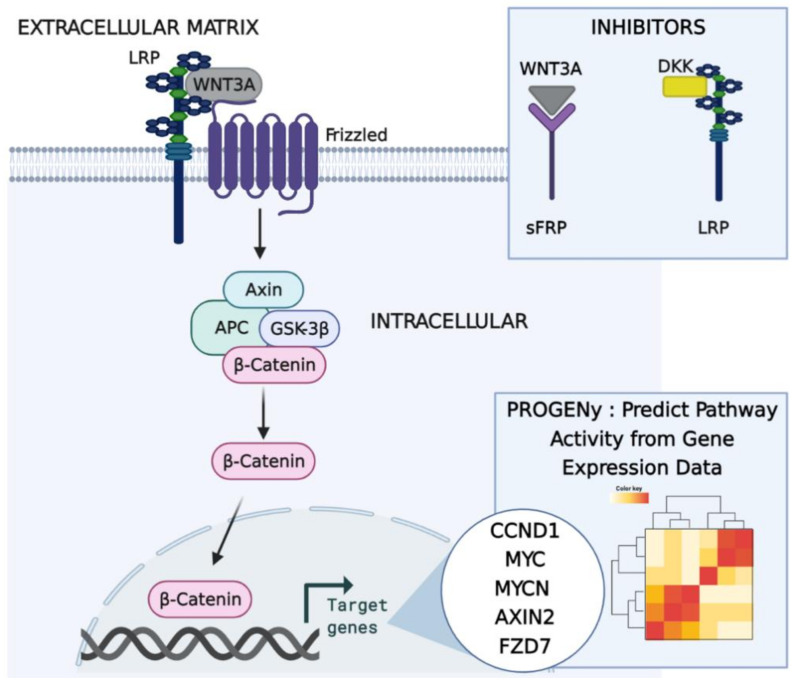
Cascade of events in the WNT Signalling Pathway. Our approach is to combine critical features (genes forming the model) from the most recent mathematical model of the WNT STN, by Kogan et al., with adjustable machine learning models [27]. The schematic represents the initial steps of activation of the WNT pathway. WNT binds to FZD (frizzled), and the co-receptor LRP5/6 (low-density lipoprotein receptor-related protein 5/6), which is recruited in the vicinity of FZD receptors upon WNT binding. This binding prevents the formation of the whole destruction complex consisting of AXIN, APC, and GSK3β (Glycogen Synthase Kinase 3 Beta), thereby causing rising levels of β-catenin, β-catenin translocation into the nucleus, and expression of WNT target genes, CCND1 (Cyclin D1), MYC, MYCN, AXIN2, and FZD7 [27]. Using machine learning tools, such as PROGENy, the WNT pathway activity can be calculated for each individual patient by analysing their gene expression [35,36]. Created with BioRender.com.

**Figure 2 ijms-22-09970-f002:**
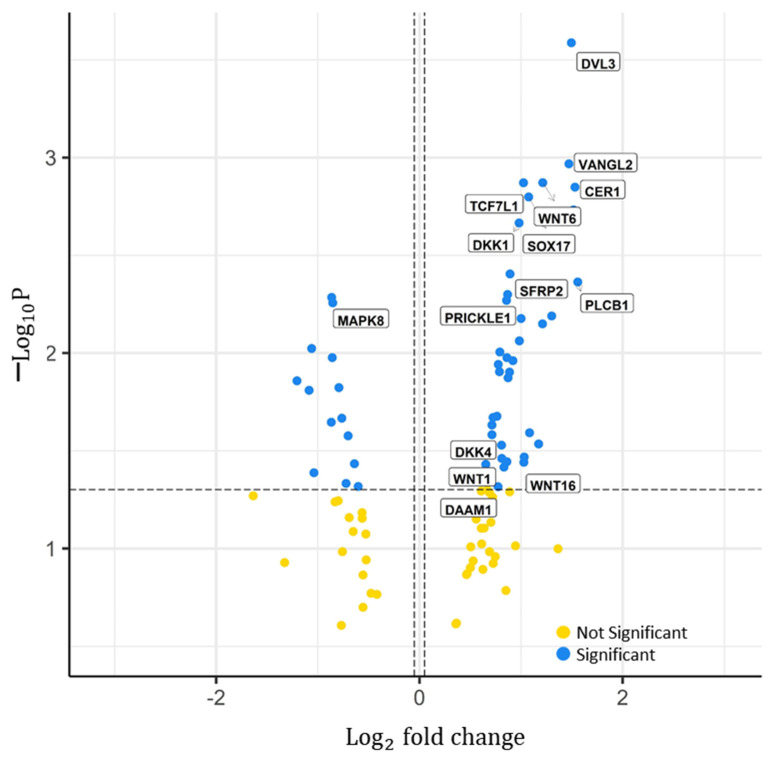
Significant genes in the WNT signalling pathway. Distribution of the *p*-value $(-\log_{10}$P) over the hazard ratio (${\mathrm{Log}}_{2}\;$ fold change) from the genes analysed from the RNA Sequencing TCGA legacy dataset. Features were divided in two; significant and not significant, represented by the blue and yellow dots, respectively. Labels represent the gene names of some significant features. *p*-value cut-off used = 0.05. ${\mathrm{Log}}_{2}\;$ fold change cut-off used = 0.5. Total number of features in WNT signalling pathway = 119. Total number of significant genes determined = 53.

**Figure 3 ijms-22-09970-f003:**
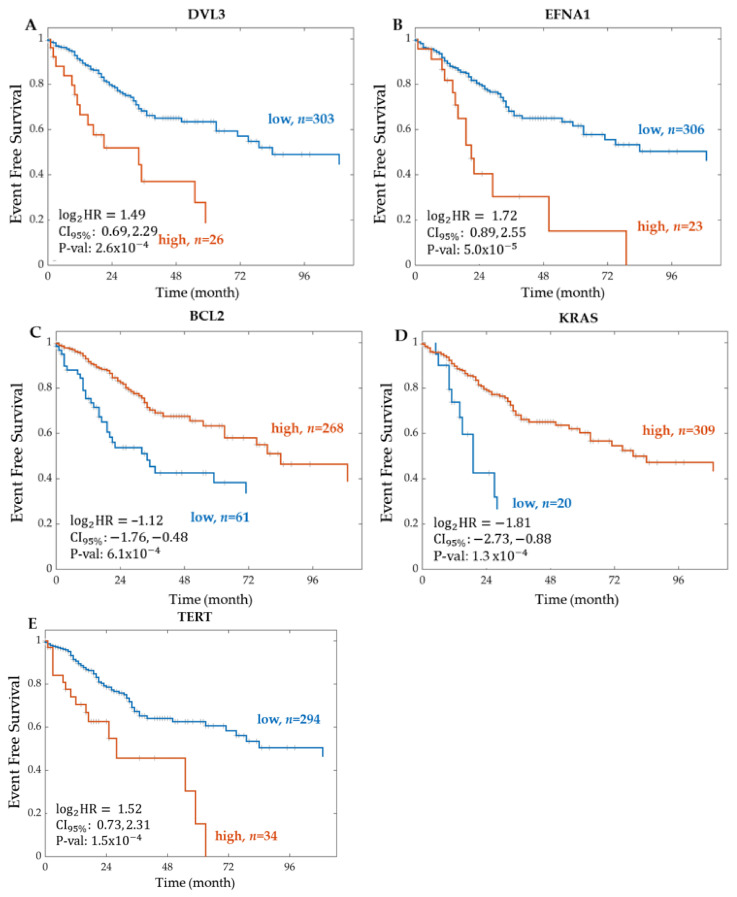
Kaplan–Meier curve of event-free survival for the most significant gene of each CRC STN. The most significant gene in each signalling pathway from the RNA Sequencing TCGA legacy dataset was (**A**) DVL3 for WNT STN, (**B**) EFNA1 for PI3K-Akt STN, (**C**) BCL2 for TP53 STN, (**D**) KRAS for MAPK STN, and (**E**) TERT for TGF-Beta STN. The patients were stratified into two groups according to the expression level of each feature. The optimal cut-off was determined using Kaplan–Meier scanning (see Methods). The groups are represented as high (orange line), and low (blue line), where *n* indicates the total number of patients in each group. For each group, the Kaplan–Meier curve of event-free survival was tested for statistically significant differences using a log-rank test. ${\mathrm{Log}}_{2}\mathrm{HR}$ = log base 2 of the hazard ratio. ${\mathrm{CI}}_{95\%}\;$ = 95% Confidence Interval. P-val = *p*-value.

**Figure 4 ijms-22-09970-f004:**
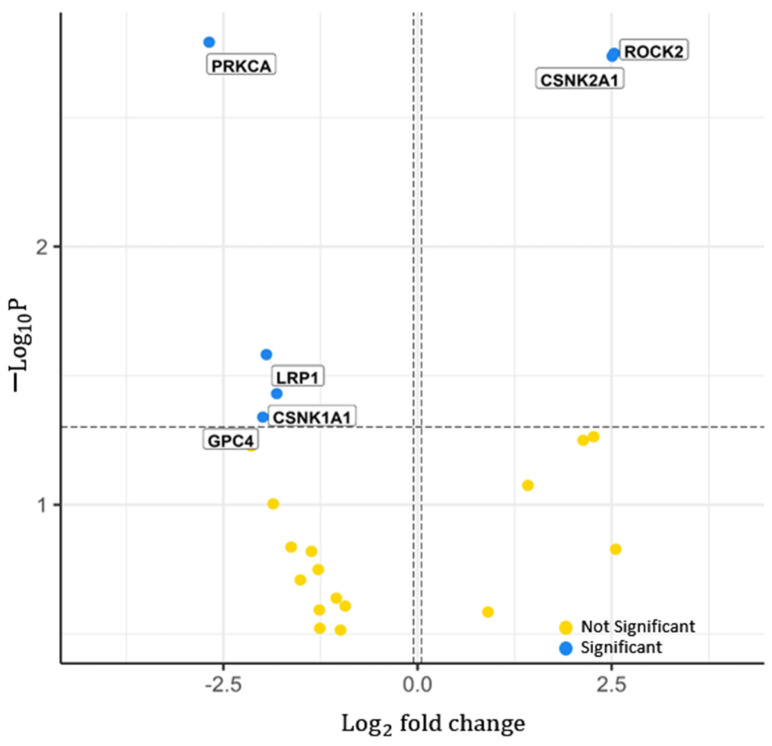
Significant proteins in the WNT signalling pathway. Distribution of the *p*-value $(-\log_{10}$P) across the hazard ratio ($\log_{2}\;$ fold change) from the proteins analysed from proteomics TCGA legacy dataset. Features were divided into two; significant and not significant, represented by the blue and yellow dots, respectively. Labels represent the gene names of some significant features. *p*-value cut-off used = 0.05. ${\mathrm{Log}}_{2}\;$ fold change cut-off used = 1. Total number of features in WNT signalling pathway = 119. Total number of significant proteomics determined = 6.

**Figure 5 ijms-22-09970-f005:**
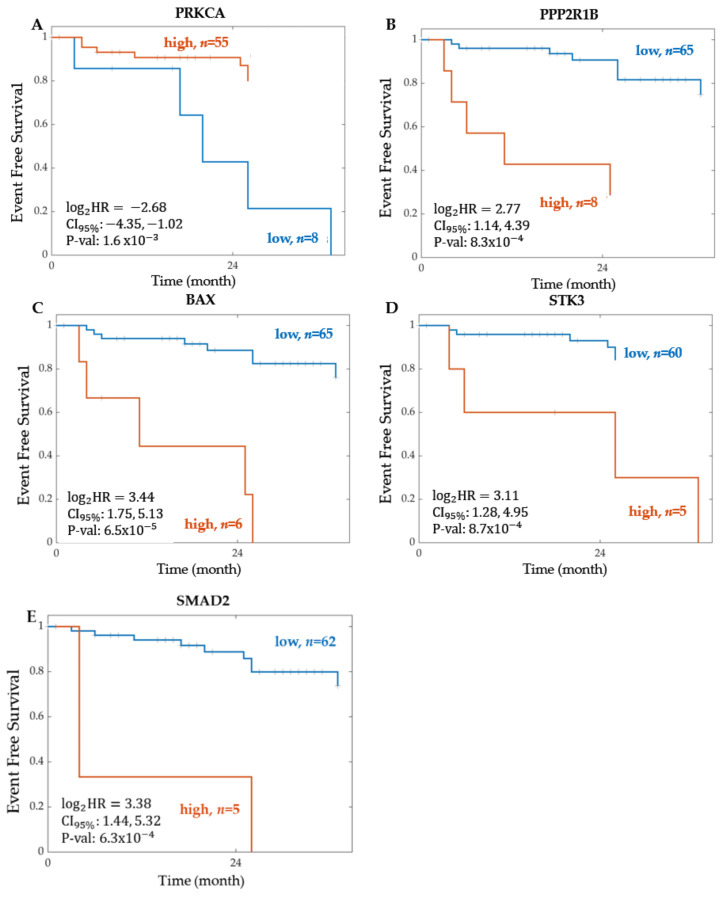
Kaplan–Meier curve of event-free survival for the most significant protein, of each CRC STN. The most significant protein in each signalling pathway from the proteomics TCGA Firehose legacy dataset was (**A**) PRKCA for WNT STN, (**B**) PPP2R1B for PI3K-Akt STN, (**C**) BAX for TP53 STN, (**D**) STK3 for MAPK STN, and (**E**) SMAD2 for TGF-Beta STN. The patients were stratified into two groups according to the expression level of each feature. The optimal cut-off was determined using Kaplan–Meier scanning (see Methods). The groups are represented as high (orange line), and low (blue line), where *n* indicates the total number of patients in each group. For each group, the Kaplan–Meier curve of event-free survival was tested for statistically significant differences using a log-rank test. ${\mathrm{Log}}_{2}\mathrm{HR}$ = log base 2 of the hazard ratio. ${\mathrm{CI}}_{95\%}\;$ = 95% Confidence Interval. P-val = *p*-value.

**Figure 6 ijms-22-09970-f006:**
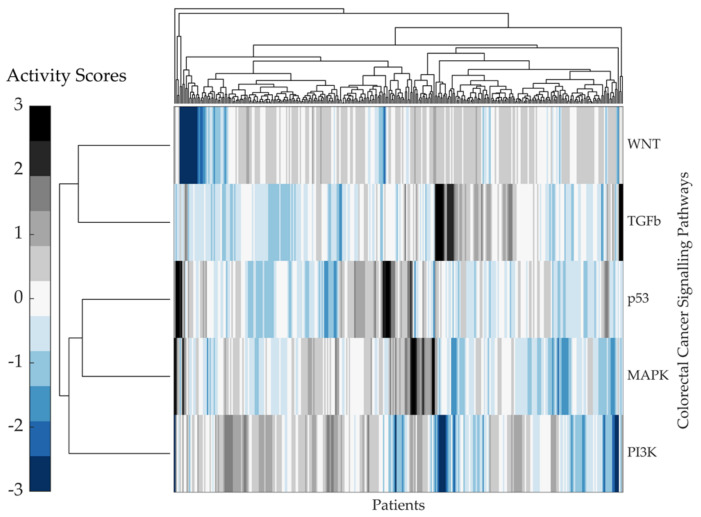
PROGENy pathway activity scores from the RNA Sequencing expression dataset. Heatmap visualising the *z*-coefficients matrix for all features in each CRC signalling pathway for all 329 patients within the RNA sequencing TCGA legacy dataset cohort. The colourmap on the left indicates the activity score values ranging from 3–−3. The Euclidean distance metric was used to pass the pairwise distance between observations between both rows and columns. The complete linkage method was used to create the hierarchical cluster tree. Data were clustered along the columns of data, then along the rows of row-clustered data.

**Figure 7 ijms-22-09970-f007:**
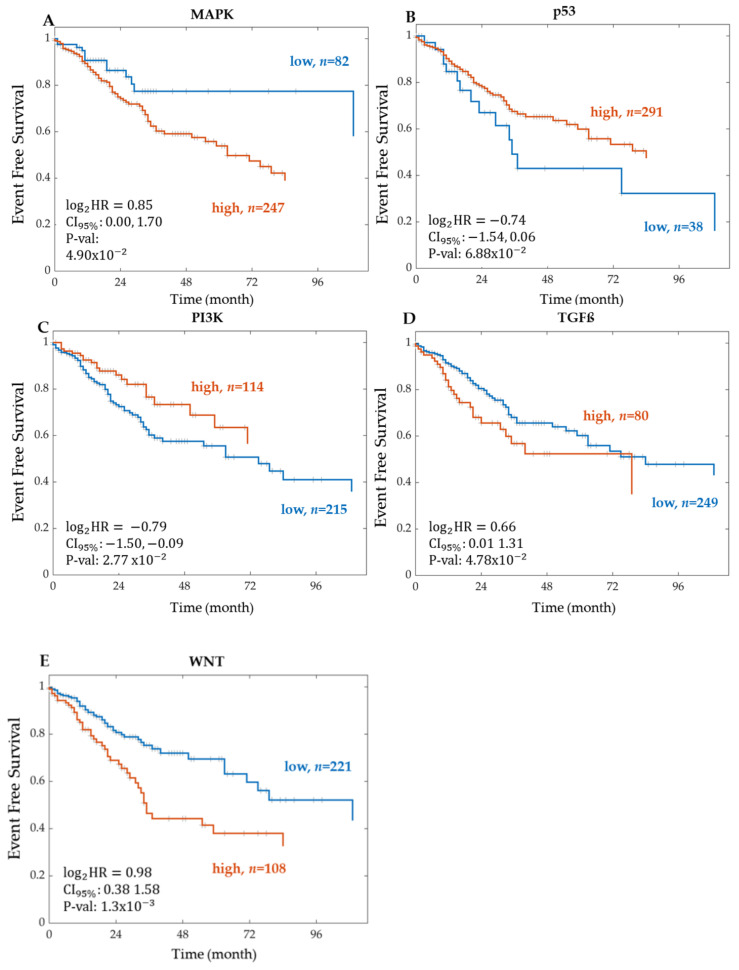
Kaplan–Meier curves of event-free survival associated with CRC PROGENy pathway activity scores. Kaplan–Meier curves associated with PROGENy pathway activity scores for the signalling pathways (**A**) MAPK, (**B**) TGFß, (**C**) PI3K, (**D**) p53, and (**E**) WNT. The patients were stratified into two groups according to the expression level of each feature. The groups are represented as high (orange line) and low (blue line), where *n* indicates the total number of patients in each group. For each group, the Kaplan–Meier curve of event-free survival was tested for statistically significant differences using a log-rank test. ${\mathrm{Log}}_{2}\mathrm{HR}$ = log base 2 of the hazard ratio. ${\mathrm{CI}}_{95\%}$ = 95% Confidence Interval. P-val = *p*-value. RNA Sequencing data is from the TCGA legacy dataset.

**Figure 8 ijms-22-09970-f008:**
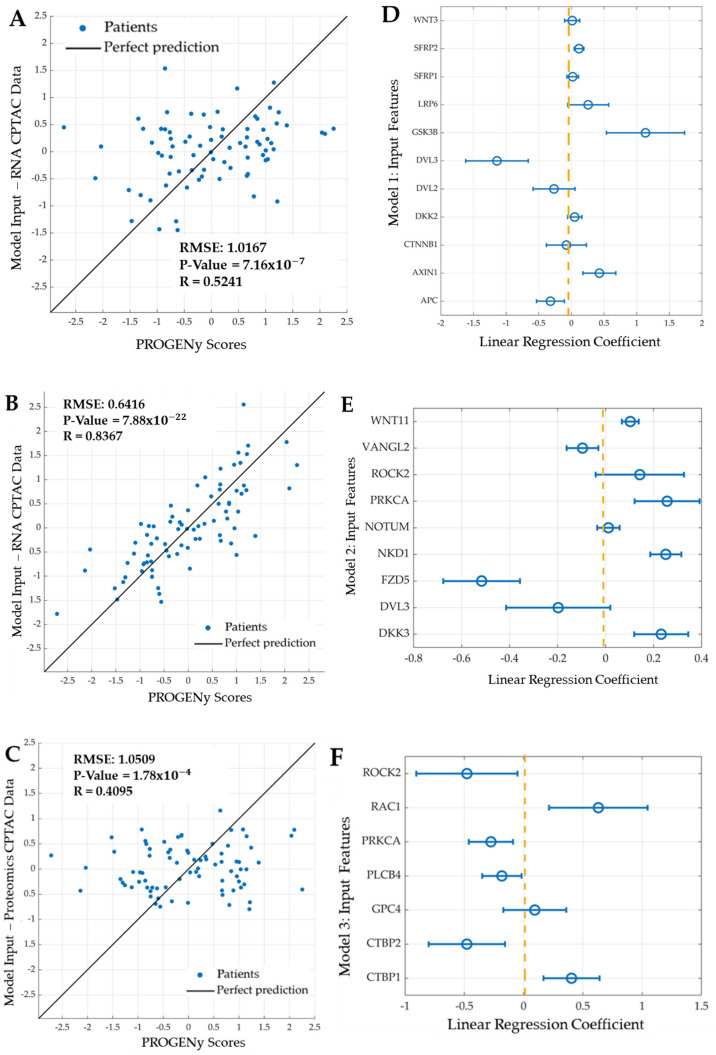
Linear regression machine learning models of the mechanistic and canonical WNT signalling pathways. Scatter plot visualising the correlation between the predicted score from (**A**) Model 1, using the 11 features of the mechanistic model (*y*-axis), and the PROGENy scores (*x*-axis). (**B**) Model 2, using the nine features of the canonical WNT model and significant genes (*y*-axis), and the PROGENy scores (*x*-axis). (**C**) Model 3, using the seven features of the canonical WNT model and significant proteins (*y*-axis), and the PROGENy scores (*x*-axis). Each blue dot represents 1 of 79 patients in the cohort. The black line indicates perfect predictions. “R” denotes the Pearson correlation coefficient, and “RMSE” is the root mean square error. The second row of figures represents the corresponding linear regression correlation coefficients for Models 1–3. Presented are the regression coefficients for the (**D**) Model 1, (**E**) Model 2, and (**F**) Model 3. The features used for each machine learning model are shown on the vertical axis, with the associated correlation coefficients shown on the horizontal axis. The blue dot represents the regression correlation coefficient, and the blue line (error bar) is the associated standard error. The dashed vertical yellow line represents the midline for the linear regression coefficients at 0.

**Figure 9 ijms-22-09970-f009:**
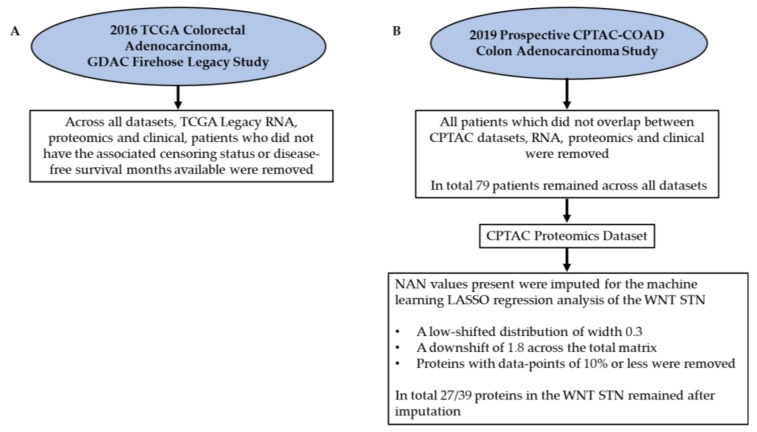
Pre-processing flow chart detailing all pre-processing steps completed prior to each analysis. Pre-processing flowchart for, (**A**) the 2016 TCGA Colorectal Adenocarcinoma, GDAC Firehose Legacy Study and (**B**) the 2019 Prospective CPTAC-COAD Colon Adenocarcinoma Study.

**Table 1 ijms-22-09970-t001:** Critical CRC STNs and the associated key functions. Nine major CRC STNs, WNT, Apoptosis, PI3K-Akt, TP53, MAPK, TGF-Beta, Cell Cycle, mTOR, and Notch are located in the first column. The “features” column represents the number of features within each STN, i.e., the number of genes in each pathway. The correlated key functions of each pathway in CRC are briefly explained in the third column [12,13,14,15,16,17,18,19,20,21,22,23].

CRC Signalling Pathways	Features	Key Functions in Colorectal Cancer
WNT	119	**Function:** Normal activation leads to tumour growth in advanced CRC. This function depends on the amount of B-catenin in the cytoplasm **Cellular Activities:** Cell fate specification, proliferation, migration, and asymmetric cell division **Activated:** WNT signal or APC mutation
PI3K/Akt	340	**Function:** Oncogenic role in the initiation and progression of CRC **Inhibition:** Reduction in CRC cell growth and increase in apoptosis **Cellular Activities:** Cell growth, proliferation, differentiation, and migration **Activated:** EGFR (epidermal growth factor receptor) Signalling Akt is a downstream effector of PI3K, mediating effects on tumour growth and progression
MAPK	252	**Function:** Oncogenic role in CRC associated with tumour growth and disease progression **Cellular Activities:** Cell growth, differentiation, and survival
TGFß	135	**Function:** Reduces colon epithelial cells proliferation and induces apoptosis and differentiation **Cellular Activities:** Cell proliferation, differentiation, migration, apoptosis, and adhesion. **Activated:** Binding of TGF-β ligands to type II TGF-β receptors Tumour suppressor in the normal epithelium Tumour promotor in the last stage of CRC
TP53	38	**Function:** Regulation of the cell cycle, DNA replication and apoptosis
Apoptosis	87	**Function:** Apoptotic cell death induction by two main pathways, intrinsic and extrinsic signalling Resulting in the formation of a death-inducing signalling complex and an apoptosome
Cell Cycle	122	**Function:** Controls cell division
mTOR	51	**Function:** Regulation of cell growth and division **Cellular Activities:** Cell growth, proliferation, and survival
Notch	48	**Function:** Promotes CRC through regulating the cell cycle and cell apoptosis by regulation of p21 and PUMA (p53 upregulated modulator of apoptosis) genes **Cellular Activities:** Normal cell development, differentiation, proliferation, and apoptosis

**Table 2 ijms-22-09970-t002:** The top two significant genes across five CRC relevant signalling pathways from the TCGA Legacy dataset. A Kaplan–Meier estimate and log-rank test were used to compute the association between patient event-free survival and gene expressions. The associated hazard ratio, 95% confidence interval, *p*-value, mafdr (estimate positive false discovery rate for multiple hypothesis testing), and standard error are apparent. “Patient” indicates the number of patients for which data were available. The last column, “significant”, indicates the number of significant genes out of the total number of genes for this pathway. The fold change and *p*-value cut-off used were 0.5 and 0.05, respectively.

Genes	Hazard Ratio	95% Confidence Interval	*p*-Value	mafdr	Standard Error	Patient	Significant
**(1) WNT**							53/119
DVL3	2.8158	(1.6159–4.9066)	2.58$\times 10^{-4}$	1.37$\times 10^{-2}$	0.2833	329	
VANGL2	2.7692	(1.5039–5.0992)	1.08$\times 10^{-3}$	1.40$\times 10^{-2}$	0.3115	329	
**(2) PI3K-Akt**							159/340
EFNA1	3.2860	(1.8496–5.8377)	4.96$\times 10^{-5}$	9.92$\times 10^{-5}$	0.2932	329	
KRAS	0.2853	(0.1502–0.5418)	1.27$\times 10^{-4}$	1.26$\times 10^{-4}$	0.3272	329	
**(3) TP53**							11/38
BCL2	0.4610	(0.2960–0.7179)	6.12$\times 10^{-4}$	0.0012	0.2260	329	
CDKN2A	1.8156	(1.1931–2.7629)	5.36$\times 10^{-3}$	0.0054	0.2142	329	
**(4) MAPK**							113/252
KRAS	0.2853	(0.1502–0.5418)	1.27$\times 10^{-4}$	2.09$\times 10^{-4}$	0.3272	329	
CACNA1I	2.7685	(1.6159–4.7432)	2.10$\times 10^{-4}$	2.09$\times 10^{-4}$	0.2747	313	
**(5) TGF-Beta**							53/135
TERT	2.8734	(1.6639–4.96201)	1.53$\times 10^{-4}$	3.05$\times 10^{-4}$	0.2787	328	
TGFB1I1	2.2381	(1.3807–3.6281)	1.08$\times 10^{-3}$	0.0011	0.2465	329	

**Table 3 ijms-22-09970-t003:** The top two significant proteins across five CRC relevant signalling pathways from the TCGA Legacy dataset. A Kaplan–Meier estimate and log-rank test were used to compute the Pearson correlation between patient event-free survival and protein expressions. The associated hazard ratio, 95% confidence interval, *p*-value, mafdr (estimate positive false discovery rate for multiple hypothesis testing), and standard error are apparent. “Patient” indicates the number of patients the corresponding protein was determined in. The last column, “significant”, in the format X/Y/Z lists the number of significant proteins in each STN (X), the number of genes within the pathway with available protein information (Y), and the total number of genes in the pathway (Z).

Genes	Hazard Ratio	95% Confidence Intervals	*p*-Value	mafdr	Standard Error	Patient	Significant
**(1) WNT**							6/27/119
PRKCA	0.1557	(0.0490–0.4947)	1.61$\times 10^{-3}$	0.0036	0.5898	63	
ROCK2	5.7810	(1.9229–17.3789)	1.78$\times 10^{-3}$	0.0036	0.5616	71	
**(2) PI3K-Akt**							29/100/340
PPP2R1B	6.8050	(2.2113–20.9407)	8.26$\times 10^{-4}$	8.32$\times 10^{-4}$	0.5735	73	
EIF4E	7.6285	(2.3168–25.1176)	8.32$\times 10^{-4}$	8.32$\times 10^{-4}$	0.6080	73	
**(3) TP53**							2/5/38
BAX	10.8648	(3.3684–35.0445)	6.54$\times 10^{-5}$	1.30$\times 10^{-4}$	0.5975	71	
BID	0.2650	(0.0814–0.8616)	2.73$\times 10^{-2}$	0.0273	0.6016	67	
**(4) MAPK**							20/80/252
STK3	8.6543	(2.4297–30.8245)	8.69$\times 10^{-4}$	0.0092	0.6481	65	
PRKCA	0.1557	(0.0490–0.4947)	1.61$\times 10^{-3}$	0.0092	0.5898	63	
**(5) TGF-Beta**							15/71/135
SMAD2	10.4343	(2.7196–40.0330)	6.30$\times 10^{-4}$	0.0013	0.6860	67	
SPTBN1	5.8318	(1.9510–17.4318)	1.60$\times 10^{-3}$	0.0016	0.5587	73	

**Table 4 ijms-22-09970-t004:** Significant genes and proteins in the WNT signalling pathway from TCGA legacy datasets. (**A**) List of significant genes and their associated hazard ratio, *p*-value, and mafdr (estimate positive false discovery rate for multiple hypothesis testing). Hazard ratios smaller than one (between 0 and 1) indicate a negative association (decreased risk), hazard ratios of >1 indicate a positive association (increased risk). The last column, proteomics, indicates whether proteomics data for this gene were available. If yes, a Y is depicted. (**B**) List of significant proteins and their associated hazard ratio, *p*-value, and mafdr. Significant features highlighted in blue were found in published mechanistic models of the WNT pathway [27,28,29,30,31,32].

A	Genes	Hazard Ratio	*p*-Value	mafdr	Patient	Proteomics
1	DVL3	2.8158	2.58$\times 10^{-4}$	1.37$\times 10^{-2}$	329	
2	VANGL2	2.7693	1.08$\times 10^{-3}$	1.40$\times 10^{-2}$	329	
3	WNT6	2.3175	1.34$\times 10^{-3}$	1.40$\times 10^{-2}$	299	
4	TCF7L1	2.0324	1.35$\times 10^{-3}$	1.40$\times 10^{-2}$	329	
5	CER1	2.8871	1.42$\times 10^{-3}$	1.40$\times 10^{-2}$	147	
6	SOX17	2.1048	1.59$\times 10^{-3}$	1.40$\times 10^{-2}$	329	
7	NKD2	2.8598	1.85$\times 10^{-3}$	1.40$\times 10^{-2}$	329	
8	DKK1	1.9732	2.16$\times 10^{-3}$	1.43$\times 10^{-2}$	309	
9	SFRP2	1.8539	3.94$\times 10^{-3}$	2.09$\times 10^{-2}$	327	Y
10	PLCB1	2.9436	4.32$\times 10^{-3}$	2.09$\times 10^{-2}$	329	
11	PRICKLE1	1.8229	5.01$\times 10^{-3}$	2.09$\times 10^{-2}$	329	
12	MAPK8	0.5491	5.19$\times 10^{-3}$	2.09$\times 10^{-2}$	329	Y
13	PRICKLE2	1.8104	5.38$\times 10^{-3}$	2.09$\times 10^{-2}$	329	
14	MYC	0.5535	5.53$\times 10^{-3}$	2.09$\times 10^{-2}$	329	
15	WIF1	2.4639	6.46$\times 10^{-3}$	2.20$\times 10^{-2}$	246	
16	CAMK2B	1.9981	6.66$\times 10^{-3}$	2.20$\times 10^{-2}$	302	
17	WNT3A	2.3137	7.09$\times 10^{-3}$	2.21$\times 10^{-2}$	243	
18	LEF1	1.9751	8.65$\times 10^{-3}$	2.52$\times 10^{-2}$	329	
19	RHOA	0.4792	9.47$\times 10^{-3}$	2.52$\times 10^{-2}$	329	Y
20	MAPK10	1.7294	9.88$\times 10^{-3}$	2.52$\times 10^{-2}$	329	
21	CAMK2D	0.5514	1.05$\times 10^{-2}$	2.52$\times 10^{-2}$	329	Y
22	WNT3	1.8141	1.06$\times 10^{-2}$	2.52$\times 10^{-2}$	328	
23	FZD8	1.8913	1.09$\times 10^{-2}$	2.52$\times 10^{-2}$	329	
24	SERPINF1	1.7116	1.14$\times 10^{-2}$	2.52$\times 10^{-2}$	329	Y
25	FZD1	1.7251	1.24$\times 10^{-2}$	2.55$\times 10^{-2}$	329	
26	NKD1	1.8480	1.25$\times 10^{-2}$	2.55$\times 10^{-2}$	329	
27	CXXC4	1.8290	2.62$\times 10^{-2}$	1.34$\times 10^{-2}$	318	
28	PPP3CA	0.4335	2.62$\times 10^{-2}$	1.39$\times 10^{-2}$	329	Y
29	PLCB3	0.5772	2.74$\times 10^{-2}$	1.50$\times 10^{-2}$	329	
30	CSNK1A1L	0.4710	2.74$\times 10^{-2}$	1.55$\times 10^{-2}$	321	
31	PLCB2	1.6955	3.46$\times 10^{-2}$	2.10$\times 10^{-2}$	329	
32	ROR2	1.6511	3.46$\times 10^{-2}$	2.14$\times 10^{-2}$	329	
33	PRKCB	0.5888	3.46$\times 10^{-2}$	2.15$\times 10^{-2}$	329	
34	WNT2	0.5479	3.52$\times 10^{-2}$	2.26$\times 10^{-2}$	329	
35	SFRP1	1.6386	3.53$\times 10^{-2}$	2.33$\times 10^{-2}$	322	
36	WNT10A	2.1170	3.69$\times 10^{-2}$	2.56$\times 10^{-2}$	328	
37	GSK3B	1.6397	3.69$\times 10^{-2}$	2.61$\times 10^{-2}$	329	Y
38	FRAT1	0.6144	3.69$\times 10^{-2}$	2.65$\times 10^{-2}$	329	
39	SENP2	2.2535	3.84$\times 10^{-2}$	2.92$\times 10^{-2}$	329	
40	DKK4	1.7497	3.84$\times 10^{-2}$	2.96$\times 10^{-2}$	301	
41	CTNNBIP1	1.5838	3.84$\times 10^{-2}$	2.97$\times 10^{-2}$	329	
42	FZD7	2.0424	4.18$\times 10^{-2}$	3.40$\times 10^{-2}$	329	
43	WNT1	1.7533	4.18$\times 10^{-2}$	3.46$\times 10^{-2}$	244	
44	DKK2	1.8152	4.18$\times 10^{-2}$	3.59$\times 10^{-2}$	326	
45	WNT16	2.0367	4.18$\times 10^{-2}$	3.62$\times 10^{-2}$	276	
46	WNT2B	0.6417	4.18$\times 10^{-2}$	3.68$\times 10^{-2}$	329	
47	NFATC4	1.5715	4.18$\times 10^{-2}$	3.70$\times 10^{-2}$	329	
48	NLK	1.7784	4.23$\times 10^{-2}$	3.83$\times 10^{-2}$	329	
49	NOTUM	1.5754	4.34$\times 10^{-2}$	4.08$\times 10^{-2}$	329	
50	CSNK2A3	0.4868	4.34$\times 10^{-2}$	4.09$\times 10^{-2}$	329	
51	WNT4	0.6060	4.82$\times 10^{-2}$	4.64$\times 10^{-2}$	329	
52	FZD5	0.6579	4.82$\times 10^{-2}$	4.81$\times 10^{-2}$	329	
53	DAAM1	1.7076	4.82$\times 10^{-2}$	4.82$\times 10^{-2}$	329	
**B**	**Proteins**	**Hazard Ratio**	* **p** * **-Value**	**mafdr**	**Patient**	
1	PRKCA	0.1557	1.62$\times 10^{-3}$	3.66$\times 10^{-3}$	63	
2	ROCK2	5.7810	1.78$\times 10^{-3}$	3.66$\times 10^{-3}$	71	
3	CSNK2A1	5.6824	1.83$\times 10^{-3}$	3.66$\times 10^{-3}$	73	
4	LRP1	0.2594	2.62$\times 10^{-2}$	3.93$\times 10^{-2}$	72	
5	CSNK1A1	0.2846	3.71$\times 10^{-2}$	4.46$\times 10^{-2}$	72	
6	GPC4	0.2512	4.58$\times 10^{-2}$	4.58$\times 10^{-2}$	50	

**Table 5 ijms-22-09970-t005:** The association between PROGENy pathway activity and patient event-free survival. PROGENy scores were analysed using Kaplan–Meier analysis and the log-rank test to obtain hazard ratios and *p*-values.

Pathway	Hazard Ratio	*p*-Value
**WNT**	1.9731	0.0013
**PI3K**	0.5775	0.0276
**TGFβ**	1.5792	0.0477
**MAPK**	1.8058	0.0489
**p53**	0.5972	0.0687

**Table 6 ijms-22-09970-t006:** Overview of the significant genes and proteins and the linear regression machine learning models. (**A**) List of the top six significant features from RNA sequencing and proteomics from TCGA Legacy datasets. (**B**) Model 1: An overview of the features used for the Mechanistic WNT pathway model. Models 2 and 3: An overview of the features used for the canonical WNT pathway, from CPTAC RNA and proteomics datasets, respectively. The number of features used and associated RMSE for each model is listed in the first column. “RMSE” = root mean squared error. Significant features are highlighted in blue.

(A) Top Six Prognostic Genes and Proteins	Features
**mRNA Sequencing** TCGA Legacy	DVL3
	VANGL2
	WNT6
	TCF7L1
	CER1
	SOX17
**Proteomics** TCGA Legacy	PRKCA
	ROCK2
	CSNK2A1
	LRP1
	CSNK1A1
	GPC4
**(B) Linear Regression Models**	**Features**
**Model 1: Mechanistic Model** Model Input: CPTAC RNA RMSE: 1.0167 11 Features	APC
	AXIN1
	CTNNB1
	DKK2
	DVL2
	DVL3
	GSK3B
	LRP6
	SFRP1
	SFRP2
	WNT3
**Model 2: Canonical WNT model with top two prognostic genes and proteins** Model Input: CPTAC RNA RMSE: 0.6416 9 Features	DKK3
	FZD5
	NKD1
	NOTUM
	WNT11
	DVL3
	PRKCA
	ROCK2
	VANGL2
**Model 3: Canonical WNT model with top two prognostic proteins** Model Input: CPTAC proteomics RMSE: 1.0509 7 Features	CTBP1
	CTBP2
	GPC4
	PLCB4
	PRKCA
	RAC1
	ROCK2

**Table 7 ijms-22-09970-t007:** Important key metrics of datasets and corresponding analyses performed for each dataset. The two studies used, TCGA Legacy and CPTAC, are located in the first column [39,40,50,51]. The “datasets” column represents the type of omic dataset used from the corresponding study in column one, with key metrics identified. The “analyses” column represents the type of analysis performed on each specified dataset. “RSEM” = RNA-sequencing by expectation-maximisation. “UQ” = upper quantile normalisation. “TMT” = tandem mass tag.

Study	Datasets	Analyses
**2016 TCGA Colorectal Adenocarcinoma, GDAC Firehose Legacy**	1. **Genomic from RNA Sequencing** 20,532 genes Cohort of 329 patients 2. **Proteomic from Mass Spectrometry** 5562 proteins Cohort of 74 patients 3. **Corresponding Clinical Dataset** Event Free Survival	1. Kaplan Meier Survival analysis for genes and proteins in pathways to find prognostic features 2. PROGENy analysis on the TCGA RNA sequencing legacy dataset to find a common core of CRC pathway activities
**2019 Prospective CPTAC-COAD Colon Adenocarcinoma**	1. **Genomic from RNA Sequencing** RNA Expression (RSEM-UQ, Log2(Val+1)) 13,482 genes Cohort of 106 patients 2. **Proteomic from Mass Spectrometry** Protein Expression (TMT, Log2ratio) 6422 proteins Cohort of 96 patients 3. **Corresponding Clinical Dataset** Cohort of 110 patients 79 patients analysed Event Free Survival	1. Determine the associations between PROGENy pathway activity scores and event free survival 2. Linear Regression Machine Learning models to predict WNT pathway activity

## Data Availability

The first dataset supporting this study were from the 2016 TCGA Colorectal Adenocarcinoma, GDAC Firehose Legacy study, previously known as TCGA provisional (cBioPortal for Cancer Genomics. Available Online: https://www.cbioportal.org/study/summary?id=coadread_tcga (accessed on 1 March 2021)) [39,40]. The second and last dataset was from the 2019 prospective CPTAC-COAD colon adenocarcinoma studies. All data were downloaded from LinkedOmics. (Datasets for Colon adenocarcinoma: Prospective CPTAC-COAD. Available Online: http://linkedomics.org/cptac-colon/ (accessed on 8 April 2021)).

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
