# Peer review of "Personalised Medicine for Colorectal Cancer Using Mechanism-Based Machine Learning Models"

_ijms, 2021, doi:10.3390/ijms22189970_

Round 1
Reviewer 1 Report
Overall: I believe the authors have addressed most of the comments by previous reviewers, but the main issue is the clarity of the manuscript and how to more proactively communicate the significance of the work. Suggest adding a few figures explaining the data, the model, why each is used, and how they come together to elevate the take-home message.
Abstract:
- You have two “ii)”, please use 1,2,3 instead for better clarity
- The two new highlighted sentences have captured the significance of this study
Introduction
- Write out the whole names for abbreviations such as PROGENy when using it for the first time.
- Add introduction about what is a mechanism-based machine learning model is and what does “adjustable” mean for linear regression? (Line 122)
- Line 110: Figure 1: Citation number for Kogan et al.
Materials and Methods:
- Please provide a summary table for your data sets with important key metrics?
- Can you create a data flow chart that including the preprocessing steps and which data is used by which analysis/ML modeling? Without this it’s really hard to understand how your analysis is conducted
- Line 738: What threshold is used to determine if a feature(genes) is significant?
Results and discussion
- Missing a section header that serves as a logical summary of the results and how each is linked together (ie why each part is important). What is the grand scheme of things? Right now it reads like a ton of facts and finding being stitched together without the main message that is being supported by each subsection.
- It is unclear to me if you used ML because it's a nice tool or because other methods have worse/not acceptable performance. What is the motivation for ML use?
- What’s the relationship between KMS analysis, PROGENy analysis, and ML modeling? Are they done in parallel or sequential? Ie each providing a piece of the puzzle or does one builds on top of another?
- Can you add subsections for the discussion section?
Reviewer 2 Report
The manuscript is now clear and sound. My only comment is about the font in the figures related to KM curves and scatter plots. Please increase the font size as they are not clearly visible.
Round 2
Reviewer 1 Report
I appreciate the author making it easy for me to review the changes. All major concerns are addressed, good for publication
This manuscript is a resubmission of an earlier submission. The following is a list of the peer review reports and author responses from that submission.
Round 1
Reviewer 1 Report
Nwaokorie et al used the lasso regression model to predict the association of STN pathways activity scores and patient event-free survival. The authors claim to improve patient stratification and help to identify potential drug targets using patient-specific mathematical and machine learning model. While the authors make efforts to address the question, the current analysis is not completely supporting the claims.
Abstract: The authors mention that mathematical and machine learning models were used, however, only machine learning models were used.
Line 34: Please specify which classifications of CRC?
Line 44-50; 114-117: Authors used five STNS known in the literature, however they mentioned 8 in table 1. There are few questions related to the chosen STNs:
- How is the choice of 5 STNs motivated?
- References 3 and 7 from where authors chose 5 STNs, also mention Notch signalling as an important pathway in CRC whereas authors did not use it and no reasoning was provided.
Line 80:82 Authors adapted the approach to combine mathematical and machine learning models, however, they did not use any mathematical model.
Figure 1: legend mentions “Combining mathematical and machine learning models of WNT signaling pathway. The figure more represent the cascade of events in WNT signaling pathway and not the mathematical/machine learning models per se. Again, which mathematical models are mentioned in the figure? Above and all this figure does not provide the message the authors seem to claim.
Line 124: How were 329 samples were selected from 631 COADREAD samples from TCGA? Please include the clinical information for the included samples. Also, please provide the link to the TCGA datasets; both transcriptomics and proteomics.
Line132: Simply fold change and p-value cannot determine the prognostic nature of the gene. The authors are looking at the important STNs which can be used as an argument, however, the current reasoning does not hold true completely for the prognostic genes from the analysis.
Line 135-Line 145: These sentences seem a part of the discussion.
Figure 3: The KM curve for one gene is shown (DVL3). Why was only this gene selected? The study seems to be only focused on the WNT pathway. The authors could have shown combined KM curves for the combined genes for each pathway.
Table 3: Did you find all 119 WNT signalling proteins in proteomics data? Similarly, for other STNs? It will be important to know if all the proteins were also recorded by the proteomics studies which is currently not presented.
Line 223: The sentence is not clear. Please reframe.
Line 267-268: “The remaining 51 prognostic features, found in this study, are not found in the mechanistic models published in the, despite their correlation to patient event-free survival.” Maybe I missed it but I could not find any correlation information in the research article?
Table 4A: 14 of the 52 genes have a hazard ratio of <1, which suggest no association of these genes with the outcome. Therefore, these genes cannot be of prognostic nature. Also leading to one of my above comments that solely fold change and p-value cannot identify the prognostic genes. They can be called significant genes.
Line294: What do you mean by anti-correlated?
Line 296-298: “Yet all pathways exhibited a wide range of activity scores that could be correlated with patient-survival.” How did the authors draw this conclusion?
Figure 6: What is the relevance of the figure? None of the pathways shows similar scores across patients? The message is not clear and does not relate with the claims of the authors in Line 296-298.
Line 327-328: “All pathways follow a similar trend, where a high pathway activation results in a shorter survival.” Is it true for all pathways? Except for MAPK and WNT, none of the other pathways holds this true.
Line 372: Model 1” Which mechanistic models are the authors talking about?
What does section 2.3.1 represent in terms of new results?
The machine learning models were built and the authors showed the correlations between the selected genes/proteins with PROGENy scores. The authors claim the prediction power of the model, however, the prediction power of the lasso models is not shown by the authors using any of the performance matrices.
Finally, the overall manuscript is written in a format of a thesis chapter rather than a research article format. Introduction, Methods, Results and discussions have intermixed sections.
Reviewer 2 Report
In the paper “Personalised Medicine for Colorectal Cancer Using Mechanistic and Machine Learning Models” by Nwaokorie&Fey the authors performed a survival analysis based on the TCGA and CPTAC datasets with a focus on pathways commonly deregulated in colorectal cancer (CRC) to identify significantly differentially expressed genes with prognostic relevance. Using PROGENy, they calculated the pathway activity scores and aimed at developing novel linear regression machine learning models to predict the contribution of individual genes to the activity scores on a patient-specific basis and gain a more comprehensive understanding of the WNT signaling pathway and its relevance in CRC.
While the approach is interesting and more comprehensive models of the WNT pathway are surely needed, the manuscript lacks important additional information and validation in its current form (for detailed information see points below). Moreover, the manuscript would largely benefit from language editing.
Major points:
Figure 1: Intracellular matrix does not exist in this context and should be labled “cytosol” or “intracellular”. Moreover, in the legend, for c-myc and n-myc the official gene names (MYC, MYCN) should be given as for the other target genes. The authors write: “WNT3A binds to LRP, and forms a receptor complex with Frizzled”. However, WNTs mainly bind to FZDs, while LRP5/6 is still considered as a co-receptor in this context which is recruited in the vicinity of FZD receptors upon WNT binding.
What was the reason for performing the study on the TCGA legacy instead of the harmonized TCGA GRCh38 dataset?
In chapter 2.1.1 the authors name the top two prognostic genes for each of the eight CRC-related pathways. However, I am missing a list depicting all identified differentially expressed genes. This is important information which should be available at least in the supplement of the paper.
In Table 2-4 I am missing the 95% CI for the given HRs. Were the p-values corrected for multiple testing (e.g. Bonferroni)? This is important when analyzing the expression of hundreds of genes in several hundred patients. Moreover, the identified prognostic genes have not been validated either experimentally, or in other datasets. Furthermore, I would recommend to perform some multivariate analysis with the identified gene versus other clinical data relevant for patient survival (e.g. tumor stage, age etc.).
In Table 3: The number of total genes in the pathways should in this case be corrected for the number of total genes with available protein information, e.g. if from 119 genes in the WNT pathway information is only available for 22 proteins, the value should be 6/22, not 6/119.
In line 262: Here, 53 genes were given as prognostic, but Table 4 only displays 52?
As a major part of the manuscript relies on the PROGENy analysis for the development of the machine learning models and here information on only 5 out of the 8 chosen CRC pathways is available, why include apoptosis, cell cycle and mTOR in the analyses above (Tab 2+3)? I would advise to leave out these three pathways in the analysis above to make the story more consistent.
In Figure 6: How do the authors explain that the PROGENy pathway activity analysis did not reveal a striking hyperactivation of the WNT pathway (most values between 0-0.5) although in most CRC patients it is the main event driving tumorigenesis?
In line 403-405 the authors mention that the LASSO-identified genes belonged to the canonical WNT pathway. Was the analysis restricted to a specific WNT subpathway, i.e. canonical or non-canonical? The term “canonical” should be explained in the introduction.
In the discussion I am missing a more detailed assessment and discussion of the results, e.g. How would you judge the identified genes/proteins from the survival analysis (Tab. 2/3), were they to be expected? Are they known activators/inhibitors of the pathways? In line 500 it is mentioned that high PRKCA levels were associated with worse event-free survival, which should be the other way around (see Fig. 5). For the discussion of the relevance of PRKCA, it should be added that it cannot only inhibit canonical WNT signaling but also enhance CRC cell death which fits to the survival data presented in Fig. 5.
Most importantly, the authors claim in their discussion that they provide patient-specific differences of (WNT) pathway activation. While the identified prognostic genes are surely interesting and definitely should be included in future mechanistic studies, the authors did not show that their machine learning method indeed correctly predicts the occurrence of events in the individual CRC patient. Instead they again compared the survival across all patients based on a high vs low group. Thus, I am missing the novelty of the approach, especially since more comprehensive pathway descriptions of WNT signaling, albeit lacking full mechanistic validation, are already available (e.g. Bayerlová et al, 2015).
Minor points:
In line 32-33 the authors state that “From the four global consensus molecular subtypes 32 (CMS) of CRC, CMS1 - 4, several of these subtypes cannot be targeted directly”. Although it is true that to date no specific drugs have been identified that target a specific CMS subtype, there are multiple studies showing that some CMS subtypes benefit from specific therapeutic regimens (Stintzing et al, 2019; Okita et al, 2018; Mooi et al, 2018). Therefore, I would advise to reformulate this sentence.
In Table 1 the source for the pathways is already indicated in the table legend, thus the column “Source” does not bring any additional information and should be removed.
Line 59-61: “STN’s can be mapped to their most frequent mutations. Interestingly, the most common gene mutation in CRC is APC which functions in the activation of the WNT STN”. This sentence is misleading as APC is no mutation but a gene and does not directly mediate the activation of the WNT pathway. Instead, APC negatively regulates WNT activity since it is an integral part of the destruction complex which targets ß-catenin for degradation. Mutations is APC thus lead to hyperactive WNT signaling.
Line 63: CRC’s should be CRCs
Line 90: WTN should be WNT
In line 137-140: “DVL proteins are key components of WNT signalling, they relay WNT signals from receptors to downstream effectors. In the canonical WNT pathway this depends on the nuclear translocation of β-catenin“. This statement is misleading: The function of DVL is not dependent on nuclear ß-catenin. Instead, WNT ligands bind to FZD receptors, and the LRP5/6 co-receptor, which leads to the recruitment of cytosolic DVL3 which then relays this signal to downstream signaling events that result in the translocation of ß-catenin to the nucleus and target gene expression.
In line 144-145 please clarify “another receptor”.
In line 266-268 parts of the sentence seem to be missing.
Line 375-378 and 393-395: Were 5 or 7 genes identified by LASSO for model 2? The numbers don’t match.
Reviewer 3 Report
Major Concerns:
- I fail to understand the supplementary file, why is it only the headers and no data?
- Many undefined abbreviations, please write their full name out the first time it is mentioned. Also there are too many abbreviations used, which lowers readability.
Abstract:
- Abstract read like a laundry list and lacks central focus, suggest rewrite.
- What’s the impact for the field?
- What’s the most important takeaway for the reader?
Introduction:
- What does “benchmark” linear regression mean exactly in line 97?
Result:
- Are the p-values significant for the field of study?
- You use 10-fold validation, but did you test your model metrics on a set of testing data that is completely unseen by the model? How well does that perform
- Can you give a more in-depth explanation of why RMSE is appropriate for this study? Also is the value acceptable for making accurate predictions? Is the predicted values within acceptable range?
- Are there outliers and if so how do you remove them?
- Linear regression:
- Have you checked if the relationship is linear?
- Are the parameters normalized?
- Are the observations independent of each other?
- Are the values normally distributed?
- Is linear regression enough to make accurate predictions? What are the caveats? Will using more complex regression methods improve prediction?
You have a very rich result and discussion section. Can you include a conclusion section that sums up the key takeaway for the paper?